# A Comprehensive Look at Maxillofacial Traumas: On the Basis of Orbital Involvement

**DOI:** 10.3390/diagnostics13223429

**Published:** 2023-11-11

**Authors:** Fatma Dilek Gokharman, Ozlem Kadirhan, Ozlem Celik Aydin, Arzu Gulsah Yalcin, Pınar Kosar, Sonay Aydin

**Affiliations:** 1Department of Radiology, SBU Ankara Education and Research Hospital, Ankara 06660, Turkey; arzugyalcinn@gmail.com (A.G.Y.); pkosar@gmail.com (P.K.); 2Department of Radiology, Erzincan University, Erzincan 24100, Turkey; ozlemkkadirhan@gmail.com (O.K.); sonay.aydin@erzincan.edu.tr (S.A.); 3Department of Pharmacology, Erzincan University, Erzincan 24100, Turkey; ozlem.celik@erzincan.edu.tr

**Keywords:** maxillofacial trauma, orbital fracture, ophthalmic complication

## Abstract

Introduction: Orbital wall fractures that may develop in maxillofacial traumas (MFTs) may cause ophthalmic complications (OCs). The aim of this study is to determine the frequency of orbital fractures (OFs) accompanying MFTs and findings suspicious for orbital traumatic involvement. Materials and Methods: Computed tomography (CT) images of 887 patients who presented to the emergency department within a 1-year period with a history of MFT were retrospectively scanned. During the examination, patients with orbital wall fractures, craniofacial bone fractures, and posttraumatic soft tissue changes were recorded. Results: OF was observed in 47 (5.3%) of the patients admitted for MFT. In cases with OFs, accompanying nasal (25.5%), ethmoid (2.1%), frontal (19.1%), maxillary (38%), and zygomatic bone fracture (10.6%), sphenoid (4.3%), and soft tissue damage (55.3%) were observed. It was observed that the pathologies mentioned at these levels were significantly higher than in patients without orbital involvement (*p* < 0.05). In our study, mild (48.9%) and moderate-severe (2.12–4.25%) OCs accompanying OFs were observed after MFT. Conclusions: The frequency of MFT varies depending on various factors, and such studies are needed to take preventive measures. Knowing the risk and frequency of orbital damage accompanying MFTs may help reduce complications by allowing rapid and accurate diagnosis.

## 1. Introduction

The maxillofacial region (MFR) consists of the upper face (frontal), mid face (maxilla, nasal complex, and zygomatic), and lower face (mandible). MFR fractures are among the most common types of trauma in trauma centers worldwide [1]. It is stated that their frequency varies significantly between countries and regions within countries due to differences in demographic, socioeconomic, and environmental factors, and their prevalence varies between 17 and 69% [2]. Male gender predominance has been reported in up to 85% of maxillofacial traumas (MFTs) worldwide [3,4].

The most common causes of MFT have been reported as road traffic accidents, assaults, falls, and sports and work-related injuries. The severity of MFT varies depending on the etiology, the magnitude and duration of the impact force, and the surface area where the impact occurs [3,5]. Injuries may be isolated or may be present as part of multiple trauma, including intracranial, ocular, and spinal, which can significantly increase the mortality or morbidity of the patient [6]. In addition to being an aesthetically important region, MFR is a condition that should be approached carefully because it contains many structures necessary for life, such as speaking, seeing, and smelling [7]. Because deficiencies in diagnosis and treatment in MFTs can cause facial deformities, chewing difficulties, visual disturbances, and even death. In this respect, emergency physicians should be careful in terms of other major traumas and their complications that may change patient management in MFTs because early assessment and intervention can significantly reduce morbidity and mortality. In the literature, the incidence of minor ocular injury associated with orbital fractures (OFs) accompanying maxillofacial fractures varies between 29.03 and 98.38%, and the incidence of major ocular injury varies between 1.7 and 20% [8,9,10]. It is important to reduce the severity of the long-term effects and avoid a condition that can have serious social and legal consequences, such as blindness. In this context, rapid diagnosis of a serious ocular injury and determining the role of ophthalmologists in the initial evaluation of facial trauma are crucial. 

This study mainly aims to define and comprehensively analyze the important anatomical and etiological aspects of MFTs with orbital involvement. 

## 2. Materials and Methods

This study was approved by a tertiary hospital Ethics Committee and was conducted in accordance with the Declaration of Helsinki. It was created by the retrospective examination of patients who were admitted to the emergency department of a tertiary hospital with a history of MFT between 1 March 2020 and 1 March 2021 and underwent non-contrast maxillofacial computed tomography (MFCT) imaging by a radiologist with 10 years of experience. Informed consent was waived due to the retrospective design of the study. A total of 887 patients were included in the study. Of the cases admitted to the emergency department due to MFT, 169 (19%) were women, and 718 (81%) were men. The ages of the cases included in the study ranged from 0 to 97 years, and the average age of the cases was 29.75 ± 19.46 years. The exclusion criteria were established to exclude instances that were not related to trauma, cases with CT scans that had motion artifacts making evaluation impossible, cases with a history of previous surgery, and cases with insufficient data in their medical history.

Non-contrast MFCTs of the patients included in the study were performed on images in the axial, coronal, and sagittal planes using standard soft tissue window and bone window settings. CT images were obtained using a Philips Brilliance 16-detector MDCT system (Philips Medical Systems, Cleveland, OH, USA) with: 120 kV; 105 mA; 0.8 mm slice thickness; 0.4 mm interval; 0.75 second rotation time; 16 × 0.75 collimations; and a 512 × 512 matrix. Data acquired in the axial plane were obtained. Coronal and sagittal images were reformatted from the axial data with a 1 mm slice thickness. All three planes were used to collect study data.

During the examination, the patients’ orbital wall (medial, lateral, inferior, superior) fractures, craniofacial bones (nasal, ethmoid, sphenoid, frontal, maxillary, mandibular, zygomatic, and temporal bones) fractures, and posttraumatic soft tissue changes, trauma type (motor vehicle accidents, attacks, falls, gunshot wounds), and orbital-related complications were recorded.

Ophthalmic complications after MFT are categorized as mild, moderate, and severe. Mild complications are periorbital edema, eyelid laceration, periorbital ecchymosis, and subconjunctival; moderate complications are enophthalmos, proptosis, restricted eye mobility, diplopia, telecanthus, dystopia, and dilated pupil; severe complications are chemosis, hyphema, perforated lens, corneal abrasion, detached retina, retrobulbar hemorrhage, traumatic optic neuropathy and blindness.

Statistical analysis: The conformity of the data to the normal distribution was examined with the Shapiro–Wilk test; numerical variables with a normal distribution were represented as mean ± standard deviation values, and categorical variables as numbers (*n*) and percentage values (%). The Student’s *t*-test was used to compare the normally distributed features in two independent groups, and the Mann–Whitney U test was used to compare the non-normally distributed features in two independent groups. Relationships between two independent variables at the categorical measurement level were tested with Pearson Chi-square or Exact Chi-square. As descriptive statistics, numbers and % values for categorical variables and mean ± standard deviation for numerical variables are given as min-max. SPSS Windows version 24.0 IBM Corporation, Armonk, NY, USA) package program was used for statistical analysis, and *p* < 0.05 was considered statistically significant.

## 3. Results

In the 1-year timeline chosen for this study, 887 cases admitted to the emergency department due to MFTs were included. Of the cases admitted to the emergency department due to MFTs, 169 (19%) were female, and 718 (81%) were male. The mean age of all subjects included in the study was 29.75 ± 19.46 years. No patient was excluded from the study.

Orbital wall fracture was observed in 47 (5.3%) patients who underwent non-contrast MFCT, while orbital wall fracture was not observed in 840 (94.7%) patients. When the mean age of patients with or without orbital wall fracture was compared, it was found that the mean age of patients with orbital wall fracture (38.98 ± 19.13) was statistically significantly higher than the mean age of patients without orbital wall fracture (29.2 ± 19.36). (*p* < 0.001). In addition, while there is no statistically significant difference in terms of age in inferior and superior wall involvements in OFs (*p* > 0.05), there is a statistically significant difference in age in medial (39.96 ± 16.2) and lateral wall (54 ± 17.18) involvements. (*p* < 0.05) (Table 1).

In our cases, OFs accompanying MFTs were most common in the inferior wall (2.9%) and the second most common in the medial wall (2.6%). The location distribution of OFs is shown in Table 2 in detail. Some orbital wall fractures of the cases in this study are shown in Figure 1, Figure 2, Figure 3 and Figure 4.

In cases with OF, accompanying nasal (*n*: 12, 25.5%), ethmoid (*n*: 1, 2.1%), frontal (*n*: 9, 19.1%), maxillary (*n*:18, 38%), sphenoid (*n*: 2, 4.3%), and zygomatic (*n*: 5, 10.6%) fractures and soft tissue damage (*n*: 26, 55.3%) were observed, and the pathologies mentioned at these levels were found to be significantly higher than in patients without orbital involvement (*p* < 0.05). There was no significant difference in association with fractures at the mandible and temporal bone levels in patients with orbital involvement compared to those without (Table 3).

MFCT images showed periorbital tissue emphysema in twenty-six cases, periorbital edema in eighteen cases, and periorbital adipose tissue (Figure 1, Figure 3, and Figure 4) and partially accompanying extraocular muscle herniation (Figure 3) in three cases. Paresthesia was observed in four cases due to fractures contacting the infraorbital canal (Figure 3 and Figure 4). In our study, mild complications were seen at rates reaching 48.9%, while the rate of moderate-severe complications varied between 2.12 and 4.25%. Mild complications, in order of frequency, were periorbital ecchymosis, periorbital edema, subconjunctival hemorrhage, and eyelid laceration. In two cases, diplopia, enophthalmia, and limitation of eye movements were observed due to fractures affecting the base level. In one case with a fracture of the medial wall, there were signs of optic neuropathy, which was thought to be due to nerve compression, in a case with a fracture involving the telecanthus and orbital canal. The distribution of complications in the cases in the study is shown in Table 4.

The main etiology was interpersonal violence-battery in both patients with MFT (*n*: 397, 44.7%) and those with orbital involvement (*n*: 21, 44.6%). (Table 5).

## 4. Discussion

Traumas have been reported as the leading cause of absenteeism and death in people under the age of 40 [11]. MFT constitutes one of the most common causes of trauma admissions worldwide. Traumas in MFR are types of injuries that require comprehensive evaluation at the time of admission, as they are close to organs, which will increase mortality and morbidity [12]. 

Factors such as geographical location, cultural differences, population density, and socioeconomic status of the country play a role in the etiology and incidence of fractures in MFTs [12]. Given the increasing prevalence of MFTs, long-term collection and analysis of epidemiological data is essential to identify and guide the implementation of traditional preventive measures. Orbital wall fractures that may develop due to MFT pose a risk in terms of ophthalmic injury due to its close proximity to the organ of vision and related muscle planes [13]. In this context, an accurate understanding of OFs and associated injuries is critical in the early diagnosis of these patients and the reduction of complications if they are treated appropriately.

The incidence of MFT and related fractures has been reported more frequently in the 20–29 age group by some authors [14,15,16,17,18,19,20,21] or in the 30–39 age group by others [22,23,24]. In our study, the mean age of the patients who underwent extraction due to maxillofacial injury was 29.75 ± 19.46 years, which was consistent with the mean age values reported in the literature. The mean age of the cases accompanied by OFs was (38.98 ± 19.13), and it was wider, although it included the age ranges specified in the literature.

In the literature, male predominance has been reported with rates ranging from 2.9:1 to 12:1, including cases accompanied by OFs in MFTs [11,14,25,26,27]. This is due to the fact that men are more prone to interpersonal conflicts than women, are involved in physical work that requires strength, and are therefore more frequently exposed to occupational accidents [22,23,24,28,29] and are more interested in extreme-contact sports [14,15,20,30,31].

In recent years, the male/female ratio tends to decrease in developed countries, where women are as much a part of society as men [32,33,34]. In our study, this rate was 4.2 in MFTs accompanied by OF, which showed that the rate of female MFT increased compared to many studies reported in the literature. Domestic violence was the most common etiology in women, with a rate of 66% (*n*: 112). In the literature, it has been reported that the frequency of maxillofacial fractures in women due to domestic violence varies between 34% and 73% [35]. Considering that partner violence is increasing, it is important for healthcare providers to identify victims, report them to relevant support units, and direct the patient to domestic violence service programs.

Similar to OFs, it has been reported that the most common etiology in all maxillofacial fractures in developed countries in recent years is interpersonal violence (59.38%), the frequency of which is increasing rather than traffic accidents (8.41%) or sports injuries (3.56%) [36]. In our study, interpersonal violence/beating (%44.7) was the most common etiology in MFT and related OFs (44.6%), and the frequency of facial trauma caused by intentional injuries can be reduced with various violence prevention programs.

In studies conducted in developing regions such as India, road traffic accidents (74.7–92.16%) were reported more frequently as the most common etiology in MFTs [29,37,38,39]. In our study, road traffic accidents (25.5%) were the second most common etiology in MFTs.

Among the causes of road traffic accidents in developing countries, reasons such as poorly defined traffic rules, inadequate traffic lighting, inadequacy of signage, and uncontrolled vehicles have been reported [30,37,38,39,40,41]. However, with the increasing population density in some developed countries, it has been reported that MFTs, due to road traffic accidents caused by drivers, driving at an earlier age, increasing carelessness and non-compliance with traffic rules [23,28,29,38,42]. Increasing the penal sanctions related to non-compliance with traffic rules in regions where traffic accidents are frequent, increasing traffic lighting systems and signs, and inspecting vehicles will reduce traffic accidents and, thus, the frequency of MFT.

Another frequent cause of MFT has been reported as falling, especially in the elderly (68–87%) and pediatric age (17–71.42%) group [32,43,44]. The reason for this situation is the presence of pathologies such as osteoporosis, in which bones become prone to fracture even in small falls in the advanced age group [45]. In the pediatric age group under 10 years of age, it is more often associated with a lack of coordination [46]. In our study, fall-related MFTs (23.2%) were in the third place.

The direction and effect of the force applied during trauma and the degree of damage to the tissue determine the location and shape of the fractures. Accordingly, the most commonly reported broken bones in MFT are mandibular and zygomatic bones. Fractures of the mandible are common because of mobility and less bone support compared to other facial bones, and it is the first bone to encounter a force trajectory during trauma or attack on the lower face. The zygomatic bone, on the other hand, has high-frequency fracture rates in MFTs due to its anatomical position and is associated with reflex lateral rotation and consequent impact, as in an attack or accident [47]. Although it is less common than mandibular and zygomatic bone, two mechanisms have been reported in OFs associated with periocular trauma and injuries: sprain and hydraulic in burst-type fractures. In the buckling theory, it has been reported that the burst fracture is caused by the force transmitted from the orbital edge. In the hydraulic theory, it is stated that the fracture is caused by a force transmitted from the eyeball [48].

In one study, the frequency of maxillofacial orbital injury was reported as 23.6% after suspected facial trauma after severe trauma. In the same study, the most frequently fractured orbital wall was the inferior wall, with a rate of 66%. The most common etiology of trauma in the patients included in this study was road traffic accidents [49]. According to a study, the formation mechanism of medial wall fractures was frequently reported as exposure to attack [50]. In our study, the frequency of orbital wall involvement was 5.3% in patients who were admitted to the hospital due to MFT, and fracture localizations were located in the inferior (2.9%) and medial wall (2.6%), respectively. In addition, the main etiology of trauma in our study was interpersonal conflict most frequently and traffic accidents less frequently. In our study, the least fractured orbital wall was the lateral and then the superior wall. In the literature, orbital superior wall fractures are the least common OF localization, with a frequency of 1–9% [48,51,52,53]. In one study, female predominance was mentioned in orbital roof fractures [50], but in our study, male predominance was found in all OFs. Complex fractures occur when two or more bones in the orbit are involved. It has been stated that medial and inferior wall fractures of the orbit have a more complex fracture pattern compared to superior and lateral fractures [48,54,55,56]. The most common complex fracture in our study was the combination of maxilla and orbital floor fracture.

In cases admitted due to maxillofacial trauma, emergency service specialists or trauma surgeons should be aware of various symptoms and physical examination findings in order not to miss possible orbital wall fractures and their associated complications. Therefore, it is important to be aware of symptoms such as periorbital pain, diplopia, blurred vision, ophthalmoplegia, or findings that may be related to eyeball displacements such as telecanthus, enophthalmia, and exophthalmos [9,57]. In these patients with suspected orbital wall fracture, CT is used as an imaging method to confirm the presence of fracture and to identify various ocular injuries and extraocular muscle compression-like findings that may be associated [55,58]. The most common radiological findings accompanying orbital wall fractures are periorbital tissue emphysema, fat herniation, and trapped muscle, respectively. The frequency of the aforementioned findings has been reported to be between 15 and 30%, and it has been reported that they do not accompany every patient with OFs [55,56,58,59,60].

In our study, periorbital tissue emphysema in twenty-six cases (55.3%), periorbital edema (38.2%) in eighteen cases, periorbital fat tissue herniation in three cases (6.3%), and paresthesia in four cases (8.5%) due to fractures contacting the infraorbital canal were observed. In the literature, 29.03–98.38% mild and 1.7–20% severe ophthalmic injuries have been reported after MFT [8,9,10]. Eyeball injuries, retrobulbar hemorrhage compressing the optic nerve, or optic neuropathy due to compression from the displaced bone are potential causes for vision loss, which are rarely reported after MFT [61,62].

Globe rupture may present with suspicious CT findings such as intraocular bleeding, lens dislocation, intraocular foreign bodies, intraocular gas, globe deformity, and wall irregularity, and its diagnosis is important because it requires surgical exploration [1]. The diagnostic sensitivity of CT in globe rupture is 67–76% [63,64].

A prevalence of 44.6% has been reported for extraocular muscle entrapment associated with diplopia, enophthalmos, and limitation of eye movements, especially associated with burst fractures of the orbital floor and medial wall [65,66]. The cause of diplopia in extraocular muscle entrapment is ischemia in the ocular muscles and then permanent damage due to Volkmann’s contracture [67]. Oculocardiac reflex, which may manifest itself with bradycardia, nausea, and vomiting, may be seen due to extraocular muscle entrapment [68]. Extraocular muscle impingement is more common in pediatric patients than in adults due to increased bone flexibility [69]. In this respect, early diagnosis and treatment are even more important, especially in pediatric patients, as it can cause fatal arrhythmias.

Due to the orbital bone frame and tight anterior soft tissue cover, it is a risky area in terms of compartment syndrome, which can cause visual loss by affecting the high pressure in the orbit and related vascular nutrition in case of posterior ciliary artery injuries within the borders of the extraocular muscle. Imaging findings indicating elevation in intraocular pressure are proptosis, elongation of the optic nerve, and globe tenting [61,70].

In our study, in accordance with the literature, mild complications (48.9%) were the most common, while moderate-severe complications (2.12–4.25%) were less common. Mild complications, in order of frequency, were periorbital ecchymosis, periorbital edema, subconjunctival hemorrhage, and eyelid laceration. In two cases, diplopia, enophthalmia, and limitation of eye movements were observed due to fractures affecting the base level. One patient with a medial wall fracture had optic neuropathy findings, which was thought to be due to nerve compression, in a patient with a fracture involving the telecanthus and orbital canal.

It is important to better understand OFs and some of the potentially serious complications associated with them, to refer the patient to the relevant branch in a short time, and to consult with them in order to reduce the devastating consequences. 

This study’s retrospective design constitutes a significant limitation. This is due to the fact that the data utilized is derived from the anamnesis information provided by the patient during the application process. Due to this situation, especially in situations such as interpersonal violence, victims may deliberately misreport the causes of trauma in order to avoid fear or some legal consequences, and in this case, misleading information about etiology may be obtained. This circumstance that we discussed might have generated bias. As part of the limitations of the study, we recommend that more samples be included, and observations be made with the participation of several centers in the future to gain a better understanding of facial fractures that cause ophthalmic injuries.

## 5. Conclusions

As a result, the frequency and localization of MFTs vary depending on the differences in demographic, socioeconomic, and environmental factors, and such studies are needed to take preventive measures. In addition, according to the data of this study, there are accompanying ophthalmic injuries in cases with orbital wall fracture due to MFT. Knowing the risk and frequency of orbital injury accompanying MFTs can increase awareness of the radiologists about the diagnosis so that prompt diagnosis and intervention can be possible and the rate of complications can reduced.

## Figures and Tables

**Figure 1 diagnostics-13-03429-f001:**
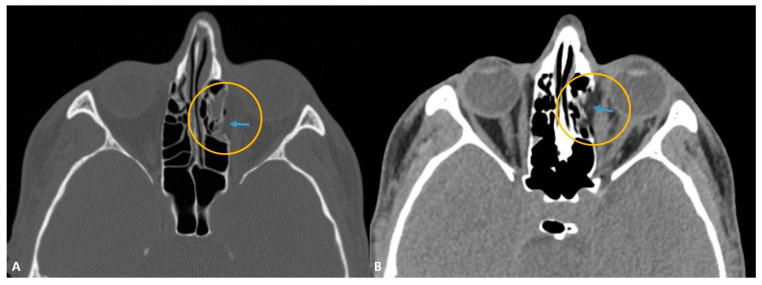
In the axial plane bone window (**A**) and parenchymal window (**B**) computed tomography (CT) images, there is a lamina papprisea fracture (yellow circle) in the medial wall of the orbit, and a partial orbital fatty tissue herniation (blue arrow) is seen at this level.

**Figure 2 diagnostics-13-03429-f002:**
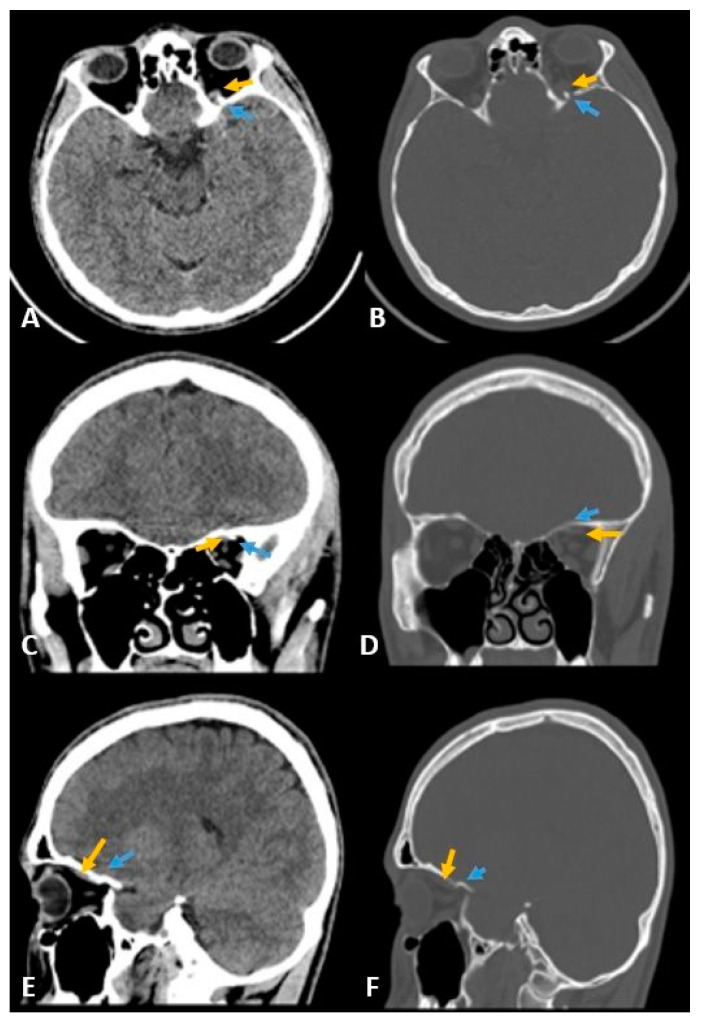
Axial (**A**), coronal (**C**), and sagittal (**E**) CT images in the parenchymal window and axial (**B**), coronal (**D**), and sagittal (**F**) CT images in the bone window show the contact of the millimetric bone fragment in the orbital roof (blue arrow) with the superior rectus muscle (yellow arrow).

**Figure 3 diagnostics-13-03429-f003:**
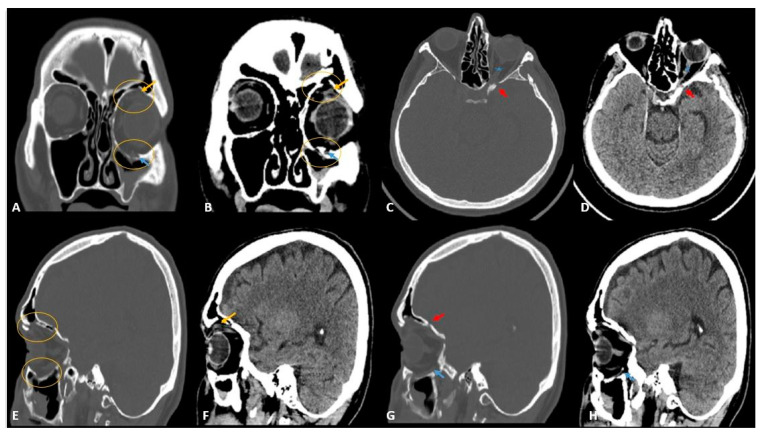
Coronal (**A**), axial (**C**), and sagittal (**E**,**G**) CT images in the bone window and coronal (**B**), axial (**D**), and sagittal (**F**,**H**) CT images in the parenchymal window. There are herniations of both fat (yellow circle) and superior rectus muscle (yellow arrow) due to the fracture in the roof of the orbit and fatty tissue (yellow circle) due to the fracture in the orbital floor. The fracture in the orbital roof extends into the optic canal (red arrow). The fracture in the lower wall also affects the infraorbital nerve (blue arrow).

**Figure 4 diagnostics-13-03429-f004:**
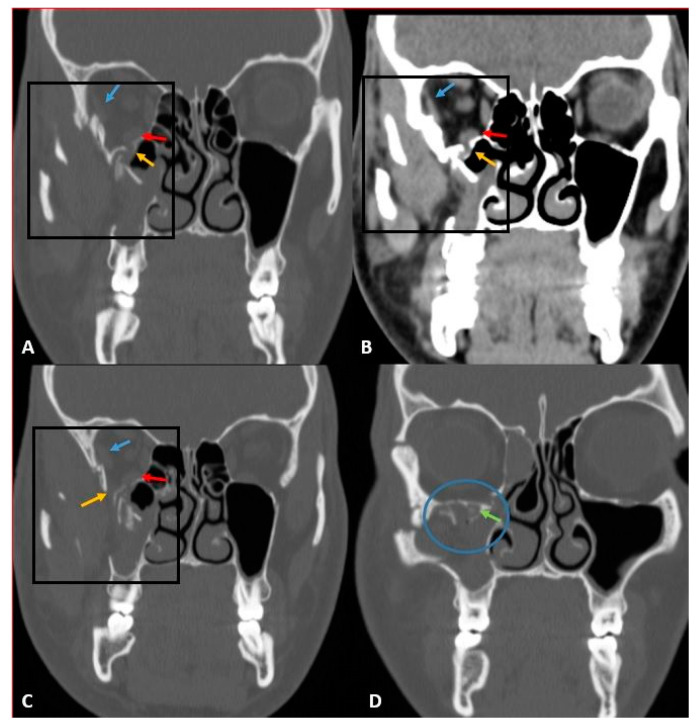
CT images in the coronal bone window (**A**,**C**,**D**) and coronal parenchyma window (**B**). There are multi-part displaced fracture lines on the orbital lateral, inferior, and maxillary sinus lateral wall (black square). Displaced bone fragments in the orbital inferior and lateral walls contact the inferior rectus (red arrow) and lateral rectus (blue arrow) muscles. Orbital fat tissue herniations (yellow arrow) are seen in different foci that herniate into the maxillary sinus due to inferior orbital wall fracture. The fracture line on the inferior orbital wall extends to the lateral wall of the infraorbital canal (blue circle), which contains the infraorbital nerve (green arrow).

**Table 1 diagnostics-13-03429-t001:** Comparison of the mean age of cases with or without orbital involvement.

	Patient Age		
	Mean ± SD	Min–Max	*p*
Orbital fracture—Yes	38.98 ± 19.13	11–87	<0.001
Orbital fracture—None	29.2 ± 19.36	0.33–97	
Medial wall fracture—Yes	39.96 ± 16.2	12–76	0.011
Medial wall fracture—None	29.48 ± 19.47	0.33–97	
Lateral wall fracture—Yes	54 ± 17.18	34–76	0.001
Lateral wall fracture—None	29.56 ± 19.36	0.33–97	
Superior wall fracture—Yes	39.09 ± 19.86	12–76	0.109
Superior wall fracture—None	29.64 ± 19.44	0.33–97	
Inferior wall fracture—Yes	36.29 ± 21.23	12–87	0.071
Inferior wall fracture—None	29.54 ± 19.37	0.33–97	

The *p*-value was obtained from the Student’s *t*-test.

**Table 2 diagnostics-13-03429-t002:** Location of orbital wall fractures.

	Location of Orbital Wall Fractures		
	Yes (*n* = 47)	None (*n* = 840)	Total (*n* = 887)
	*n* (%)	*n* (%)	*n* (%)
Medial wall fracture—Yes	23/47 (48.9)	0 (0)	23/887 (2.6)
Medial wall fracture—None	24/47 (51.1)	840/840 (100)	864/887 (97.4)
Lateral wall fracture—Yes	8/47 (17)	0 (0)	8/887 (0.9)
Lateral wall fracture—None	39/47 (83)	840/840 (100)	879/887 (99.1)
Superior wall fracture—Yes	11/47 (23.4)	0 (0)	11/887 (1.2)
Superior wall fracture—None	36/47 (76.6)	840/840 (100)	876/887 (98.8)
Inferior wall fracture—Yes	26/47 (55.3)	3/840 (0.4)	29/887 (3.3)
Inferior wall fracture—None	21 /47 (44.7)	837/840 (99.6)	858/887 (96.7)

**Table 3 diagnostics-13-03429-t003:** Other maxillofacial region fracture locations and soft tissue involvement in cases with and without orbital involvement.

	Orbital Involvement			
	Yes (*n* = 47)	None (*n* = 840)	Total (*n* = 887)	
	*n* (%)	*n* (%)	*n* (%)	*p*
Nasal bone fracture—Yes	12/47 (25.5)	121/840(14.4)	133/887 (15)	<0.001
Nazal bone fracture—None	35/47 (74.5)	719/840 (85.6)	754/887 (85)	
Soft tissue involvement—Yes	26/47 (55.3)	26/840 (3.1)	52/887 (5.9)	<0.001
Soft tissue involvement—None	21/47 (44.7)	814/840 (96.9)	835/887 (94.1)	
Ethmoid bone fracture—Yes	1/47 (2.1)	0/840 (0)	1/887 (0.1)	<0.001
Ethmoid bone fracture—None	46/47 (97.9)	840 /840 (100)	886/887 (99.9)	
Sphenoid bone fracture—Yes	2/47 (4.3)	0/840 (0)	2/887 (0.2)	0.003
Sphenoid bone fracture—None	45/47 (95.7)	840/840 (100)	885/887 (99.8)	
Frontal bone fracture—Yes	9/47 (19.1)	3/840 (0.4)	12/887 (1.4)	<0.001
Frontal bone fracture—None	38/47 (80.9)	837/840 (99.6)	875/887 (98.6)	
Maxillary bone fracture—Yes	18/47 (38.3)	5/840 (0.6)	23/887 (2.6)	<0.001
Maxillary bone fracture—None	29/47 (61.7)	835/840 (99.4)	864/887 (97.4)	
Mandibula bone fracture—Yes	2/47 (4.3)	5/840 (0.6)	7/887 (0.8)	0.059
Mandibula bone fracture—None	45/47 (95.7)	835/840 (99.4)	880/887 (99.2)	
Zygomatic bone fracture—Yes	5/47 (10.6)	2/840 (0.2)	7/887 (0.8)	<0.001
Zygomatic bone fracture—None	42/47 (89.4)	838/840 (99.8)	880/887 (99.2)	
Temporal bone fracture—Yes	1/47 (2.1)	0/840 (0)	1/887 (0.1)	0.053
Temporal bone fracture—None	46/47 (97.9)	840/840 (100)	886/887 (99.9)	

The *p*-value was obtained from the Chi-square or Fisher Exact test.

**Table 4 diagnostics-13-03429-t004:** Ophthalmic complication severity and distribution of related complications in cases with orbital wall fracture.

Ophthalmic Complication	Number (*n*)	Percentage (%)
**MILD**		
Periorbital edema	18	38.29
Eyelid laceration	6	7.83
Periorbital ecchymosis	23	48.93
Subconjunctival hemorrhage	14	29.78
**MODERATE**		
Enophthalmos	2	4.25
Restriction of eye muscles	2	4.25
Diplopia	2	4.25
Telecanthus	1	2.12
Dystopia	1	2.12
Dilated pupil	1	2.12
**SEVERE**		
Corneal abrasion	1	2.12
Traumatic optical neuropathy	1	2.12

**Table 5 diagnostics-13-03429-t005:** Comparison of the distribution of extraction reasons in cases with and without orbital involvement.

Reason for MFCT Shooting	Orbital Involvement		MFT
	Yes (*n*: 47)	None (*n*: 840)	Total (*n*: 887)
	*n* (%)	*n* (%)	(%)
**Interpersonal violence/beaten**	21/47 (44.6)	376/840 (44.7)	(44.7)
**Traffic accident**	17/47 (36.1)	210/840 (25)	(25.5)
**Fall**	9/47 (19.1)	197/840 (23.4)	(23.2)
**Work accident**	0 (0)	32/840 (3.8)	(3.6)
**Hitting**	0 (0)	25/840 (2.9)	(2.8)

Maxillofacial computed tomography (MFCT), maxillofacial trauma (MFT).

## Data Availability

The data presented in this study are available on request from the corresponding author.

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
