# Peer review of "A Comprehensive Look at Maxillofacial Traumas: On the Basis of Orbital Involvement"

_diagnostics, 2023, doi:10.3390/diagnostics13223429_

Round 1
Reviewer 1 Report
Comments and Suggestions for Authors
Dear authors,
Overall, the manuscript titled "A Comprehensive Look at Maxillofacial Traumas" provides valuable insights into the frequency and associated complications of maxillofacial traumas (MFTs), with a particular focus on orbital fractures. Here are my comments and suggestions for improvement:
1. I would suggest a more specific title to your study, the current title is vague and does not really reflect the exact aims of the study
2. The abstract should provide a more concise summary of the key findings, methods, and implications of the study. It is quite lengthy and detailed for an abstract.
3. The introduction provides a good background and context for the study. However, it would benefit from a clear statement of the research objectives or hypotheses.
4. In the material and methods section, please provide more information about the study population, such as age range, gender distribution, and any relevant demographics. Clarify whether there were any exclusion criteria beyond missing data. Mention the specific CT imaging protocols used in the study. Include a brief description of the statistical methods used for data analysis.
5. The results section is well-organized and presents the findings clearly. Clarify whether the percentages provided are based on the total number of patients (887) or the subset with orbital fractures (47). When discussing the frequency of orbital wall fractures, it may be helpful to compare the findings with existing literature.
6. The discussion should include a more comprehensive review of the relevant literature to support the findings and interpretations. When discussing the etiology of maxillofacial trauma, consider providing statistics or references to support the claims made. Explain the clinical significance of the findings in the context of patient care and outcomes. Address the limitations of the study in more detail, including potential sources of bias or confounding factors. Discuss the implications of the study's findings for clinical practice and future research.
7. Minor English refinement is advised.
Overall, the manuscript presents valuable research on maxillofacial traumas and their associated complications. To enhance its clarity and readability, I recommend addressing the points mentioned above and ensuring that the abstract provides a concise summary of the study's key findings and implications.
Comments on the Quality of English LanguageMinor English refinement is advised.
Author Response
Manuscript ID:diagnostics-2643303
Manuscript Title :A COMPREHENSIVE LOOK AT MAXILLOFACIAL TRAUMAS
Dear Editor,
Thank you for giving us the opportunity to submit a revised draft of the manuscript. We appreciate the time and effort that you and the reviewers dedicated to providing feedback on our manuscript and are grateful for the insightful comments on and valuable improvements to our paper. We have incorporated most of the suggestions made by the reviewers. These changes were highlighted by the trace changes function through the manuscript and are also mentioned in order below.
1.I would suggest a more specific title to your study, the current title is vague and does not really reflect the exact aims of the study
Response: The title of the article has been changed to ‘A COMPREHENSIVE LOOK AT MAXILLOFACIAL TRAUMAS:ON THE BASIS OF ORBITAL INVOLVEMENT '.
2.The abstract should provide a more concise summary of the key findings, methods, and implications of the study. It is quite lengthy and detailed for an abstract.
Response: The abstract of the article has been shortened:
‘Abstract: Background:Orbital wall fractures that may develop in maxillofacial traumas (MFTs) may cause ophthalmic complications.The aim of this study is to determine the frequency of orbital fractures accompanying MFTs and findings suspicious for orbital traumatic involvement.Methods:Computed tomography (CT) images of 887 patients who presented to the emergency department within a 1-year period with a history of MFT were retrospectively scanned. During the examination, patients with orbital wall fractures, craniofacial bones fractures and posttraumatic soft tissue changes were recorded.Results: Orbital fracture was observed in 47 (5.3%) of the patients admitted for MFT.In cases with orbital fractures, accompanying nasal (25.5%), ethmoid (2.1%), frontal (19.1%), maxillary (38%) , and zygomatic bone fracture (10.6%), sphenoid (4.3 %) and soft tissue damage (55.3%) were observed.It was observed that the pathologies mentioned at these levels were significantly higher than in patients without orbital involvement (p<0.05).In our study, mild (48.9%) and moderate-severe (2.12-4.25%) ophthalmic complications accompanying orbital fractures were observed after MFT. Conclusions:The frequency of MFT varies depending on various factors, and such studies are needed to take preventive measures.Knowing the risk and frequency of orbital damage accompanying MFTs may help reduce complications by allowing rapid and accurate diagnosis.’
3.The introduction provides a good background and context for the study. However, it would benefit from a clear statement of the research objectives or hypotheses.
Response :The aim was redefined as to be more precise and concise. 'This study aims to identify and comprehensively analyze important anatomical and etiological aspects of MFTs with mainly orbital involvement'.
4.In the material and methods section, please provide more information about the study population, such as age range, gender distribution, and any relevant demographics. Clarify whether there were any exclusion criteria beyond missing data. Mention the specific CT imaging protocols used in the study. Include a brief description of the statistical methods used for data analysis.
Response: The material and method section was rearranged based on the aforementioned suggestions.
- The results section is well-organized and presents the findings clearly. Clarify whether the percentages provided are based on the total number of patients (887) or the subset with orbital fractures (47). When discussing the frequency of orbital wall fractures, it may be helpful to compare the findings with existing literature.
Response : Corrections have been made in the relevant tables.
6.The discussion should include a more comprehensive review of the relevant literature to support the findings and interpretations. When discussing the etiology of maxillofacial trauma, consider providing statistics or references to support the claims made. Explain the clinical significance of the findings in the context of patient care and outcomes. Address the limitations of the study in more detail, including potential sources of bias or confounding factors. Discuss the implications of the study's findings for clinical practice and future research.
Response : The discussion section has been rearranged according to your suggestions.

Reviewer 2 Report
Comments and Suggestions for Authors
I believe this was an attempt to comprehensively evaluate the effects of maxillofacial trauma depending on various factors. This attempt remains a missed opportunity, and the initial intent was not pursued throughout, authors narrow their focus on the orbital fractures accompanying MFTs and findings suspicious for orbital traumatic involvement. As such, it adds merely any glimpse.
Firstly, the authors should strictly adhere to the instructions to authors, and they are advised to check recently published articles in this journal. This accounts for both structure and outlook of the abstract and the use of abbreviations. Acronyms/Abbreviations should be defined the first time they appear in each of three sections: the abstract, the main text; the first figure or table. When defined for the first time, the acronym/abbreviation should be added in parentheses after the written-out form, any subsequent use should be CONSECUTIVE, for instance – if “maxillofacial region” is abbreviated MFB as in ln. 31; each next time You use it, it should mean exactly “maxillofacial region”.
References must be updated (only a few of them are allowed to be older than 5 yrs).
Comments on the Quality of English Languagetext should be edited throughout.
Author Response
Manuscript ID:diagnostics-2643303
Manuscript Title :A COMPREHENSIVE LOOK AT MAXILLOFACIAL TRAUMAS
Dear Editor,
Thank you for giving us the opportunity to submit a revised draft of the manuscript. We appreciate the time and effort that you and the reviewers dedicated to providing feedback on our manuscript and are grateful for the insightful comments on and valuable improvements to our paper. We have incorporated most of the suggestions made by the reviewers. These changes were highlighted by the trace changes function through the manuscript and are also mentioned in order below.
Firstly, the authors should strictly adhere to the instructions to authors, and they are advised to check recently published articles in this journal. This accounts for both structure and outlook of the abstract and the use of abbreviations. Acronyms/Abbreviations should be defined the first time they appear in each of three sections: the abstract, the main text; the first figure or table. When defined for the first time, the acronym/abbreviation should be added in parentheses after the written-out form, any subsequent use should be CONSECUTIVE, for instance – if “maxillofacial region” is abbreviated MFB as in ln. 31; each next time You use it, it should mean exactly “maxillofacial region”.
Response: Corrections were made in the article by paying attention to abbreviations.
References must be updated (only a few of them are allowed to be older than 5 yrs).
Response: Some of the references have been updated. However, references containing information about numerical data in the literature have not been changed.
- Arangio P, Vellone V, Torre U, et al. Maxillofacial fractures in the province of Latina, Lazio, Italy: review of 400 injuries and 83 cases. 2014;42(5):583-7.
- Abhinandan Patel K, Sneha T, Reddy KR, et al. Changing trends in the pattern of maxillofacial injuries in helmeted motorcycle accident patients when compared to non-helmeted motorcycle accident patients. 2023:1-7.
- Gupta P, Bansal S, Sinwar PD, et al. A Retrospective Study of Maxillofacial Fractures at a Tertiary Care Centre in North India: A Review of 1674 Cases. 2023:1-5.
- Shivakotee S, Menon S, Sham M, et al. Midface fracture pattern in a tertiary care hospital–A prospective study. 2022;13(2):238-42.
- Sharifi F, Samieirad S, Grillo R, et al. The causes and prevalence of maxillofacial fractures in Iran: a systematic review. 2023;12(1):1.
- Boffano P, Roccia F, Zavattero E, et al. European Maxillofacial Trauma (EURMAT) project: a multicentre and prospective study. 2015;43(1):62-70.
37.Adam S, Sama HD, Akpoto YM, et al. Orbital Floor Fractures: Epidemiological, Clinical and Therapeutical Study at Sylvanus Olympio University Teaching Hospital in Lomé about 51 Cases. 2021;11(9):373-86.
- Hoonpongsimanont W, Ghanem G, Saadat S, et al. Correlation between alcohol use disorders, blood alcohol content, and length of stay in trauma patients. 2021;14(1):42.
- Kambalimath H, Agarwal S, Kambalimath D, et al. Maxillofacial injuries in children: a 10-year retrospective study. J Maxillofac Oral Surg. 2013; 12: 140-44.
- Harrison P, Hafeji S, Green AO, et al. Isolated paediatric orbital fractures: a case series and review of management at a major trauma centre in the UK. 2023;27(2):227-33.
- Toyohara Y, Mito N, Nakagawa S, et al. Asystole Due to Oculocardiac Reflex during Surgical Repair of an Orbital Blowout Fracture. 2022;10(9).
- Al-Qattan MM, Al-Qattan YMJP, Open RSG. “Trap door” orbital floor fractures in adults: are they different from pediatric fractures? 2021;9(4).
- Goelz L, Syperek A, Heske S, et al. Retrospective cohort study of frequency and patterns of orbital injuries on whole-body CT with maxillofacial multi-slice CT. 2021;7(3):373-86.

Reviewer 3 Report
Comments and Suggestions for Authors
The study is about the frequency of orbital fractures secondary to maxillofacial traumas and the incidence of different anatomical locations of orbital wall fractures.
The manuscript is well written, however, the study only reported frequency and descriptive analysis.
Introduction:
Page 2, line 56-59. This study aimed to determine the frequency of orbital involvement accompanying MFTs, the incidence of different anatomical locations of orbital wall fractures (medial, lateral, inferior and superior walls), findings suspicious for orbital traumatic involvement and the etiology of injury.
Incomplete sentence. Finding suspicious?
Materials and Methods:
Page 2, line 63. It was created by the retrospective examination of patients who applied to the emergency department of a tertiary hospital.
Applied to? Should be admitted to….
Page 2, line 66. Informed consent was waived due to the retrospective design of the study. However, in Page 10, line 324: Informed Consent Statement: Informed consent was obtained from all subjects involved in the study.
Which one is true?
Page 2, line 83. The conformity of the data to the normal distribution was examined with the Shaphiro Wilk test. There is a dash, Shapiro-wilk. The results did not mention whether the data was normally distributed or not.
Results:
Page 3, line 105. The decimal points were used - significant difference in age in medial (39.96±16.2) and lateral wall (54±17.18) involvements. However, Table 1 used comma instead of decimal points e.g. 38,98 etc. Please use decimal points in Table 1.
Page 2, Table 2 – Please use decimal points.
Page 6, Table 3 - Please use decimal points.
Page 6, line 139. maxillary (n:18, 38%) ,3). What do you mean by no. 3 outside the bracket?
Table 3 – Sfenoid bone fracture. The p value is 0.003 which is <0.05, means the result was significant but reported as no significant difference in line 142.
Table 3. The table legend stated Ki square or Fisher Exact test. What is Ki square?
Page 7, Table 4. Periorbital ödem, Göz kapağı laserasyonu, Periorbital ekimoz, Subkonjunktival hemoraji. Please use English.
Page 7, Table 4. Spacing errors – the Number (N) and Percentage (%) of the results under MODERATE as well as SEVERE (including spacing for Corneal abrasion).
Page 7, Table 5. Maksillofasiyal bilgisayarlı tomografi (MFCT),maksillofasiyal travma (MFT). Please change the table legend to English.
Page 7, Table 5. What do you mean by Hit?
Page 7, Table 5. Please use decimal points instead of comma.
Page 8, line 171. What do you mean by MFB?
Conclusions:
Page 10, line 319. Knowing the risk and frequency of orbital injury accompanying MFTs may help to reduce complications by enabling rapid and accurate diagnosis.
How?
Comments on the Quality of English LanguageModerate English editing needed
Author Response
Manuscript ID:diagnostics-2643303
Manuscript Title :A COMPREHENSIVE LOOK AT MAXILLOFACIAL TRAUMAS
Dear Editor,
Thank you for giving us the opportunity to submit a revised draft of the manuscript. We appreciate the time and effort that you and the reviewers dedicated to providing feedback on our manuscript and are grateful for the insightful comments on and valuable improvements to our paper. We have incorporated most of the suggestions made by the reviewers. These changes were highlighted by the trace changes function through the manuscript and are also mentioned in order below.
Introduction:
Page 2, line 56-59. This study aimed to determine the frequency of orbital involvement accompanying MFTs, the incidence of different anatomical locations of orbital wall fractures (medial, lateral, inferior and superior walls), findings suspicious for orbital traumatic involvement and the etiology of injury.
Incomplete sentence. Finding suspicious?
Response : The relevant sentence has been changed to : 'This study aims to identify and comprehensively analyze important anatomical and etiological aspects of MFTs with mainly orbital involvement'.
Materials and Methods:
Page 2, line 63. It was created by the retrospective examination of patients who applied to the emergency department of a tertiary hospital.
Applied to? Should be admitted to….
Response :’ It was created by the retrospective examination of patients who admitted to the emergency department of a tertiary hospital with a history of MFT between March 1, 2020 and March 1, 2021 and underwent non-contrast maxillofacial computed tomography (MFCT) imaging by a radiologist with 10 years of experience’.
Page 2, line 66. Informed consent was waived due to the retrospective design of the study. However, in Page 10, line 324: Informed Consent Statement: Informed consent was obtained from all subjects involved in the study.
Which one is true?
Response : Informed consent was waived due to the retrospective design of the study. However, written consent was obtained for the publication of the images of the cases included in the study after anonymization.
Page 2, line 83. The conformity of the data to the normal distribution was examined with the Shaphiro Wilk test. There is a dash, Shapiro-wilk. The results did not mention whether the data was normally distributed or not.
Response : The relevant sentence has been changed and descriptive information has been added :’The conformity of the data to the normal distribution was examined with the Shaphiro -Wilk test, numerical variables with a normal distribution were represented as mean ± standard deviation values, and categorical variables as numbers (n) and percentage values (%)’.
Results:
Page 3, line 105. The decimal points were used - significant difference in age in medial (39.96±16.2) and lateral wall (54±17.18) involvements. However, Table 1 used comma instead of decimal points e.g. 38,98 etc. Please use decimal points in Table 1.
Page 2, Table 2 – Please use decimal points.
Page 6, Table 3 - Please use decimal points.
Response : Decimal numbers have been changed.
Page 6, line 139. maxillary (n:18, 38%) ,3). What do you mean by no. 3 outside the bracket?
Response : The relevant sentence has been changed
Table 3 – Sfenoid bone fracture. The p value is 0.003 which is <0.05, means the result was significant but reported as no significant difference in line 142.
Response : Corrections were made in the relevant sentence.
Table 3. The table legend stated Ki square or Fisher Exact test. What is Ki square?
Response : A spelling error in the relevant sentence has been corrected.
Page 7, Table 4. Periorbital ödem, Göz kapağı laserasyonu, Periorbital ekimoz, Subkonjunktival hemoraji. Please use English.
Response : Related translation errors have been corrected.
Page 7, Table 4. Spacing errors – the Number (N) and Percentage (%) of the results under MODERATE as well as SEVERE (including spacing for Corneal abrasion).
Response : Corrections were made in the relevant section.
Page 7, Table 5. Maksillofasiyal bilgisayarlı tomografi (MFCT),maksillofasiyal travma (MFT). Please change the table legend to English.
Response : Translation errors have been corrected in the relevant section.
Page 7, Table 5. What do you mean by Hit?
Response : It is stated that the maxillofacial region hit something.
Page 7, Table 5. Please use decimal points instead of comma.
Response : A correction has been made in the relevant table.
Page 8, line 171. What do you mean by MFB?
Response : The maxillofacial region (MFB) consists of the upper face (frontal), mid face (maxilla, nasal complex, and zygomatic), and lower face (mandible).
Conclusions:
Page 10, line 319. Knowing the risk and frequency of orbital injury accompanying MFTs may help to reduce complications by enabling rapid and accurate diagnosis.
How?
Response: The relevant sentence has been changed as follows : ‘Knowing the risk and frequency of orbital injury accompanying MFTs can increase awareness of the radiologists about the diagnosis, so that prompt diagnosis and intervation can be possible , the rate of complications can reduce’ .

Round 2
Reviewer 2 Report
Comments and Suggestions for Authors
Congratulations on excellent paper!
Comments on the Quality of English LanguageGrammar should be checked by someone skilled!
Reviewer 3 Report
Comments and Suggestions for Authors
The authors have addressed all my comments for this paper and answered the questions accordingly. The paper has been significantly improved after revising.
Comments on the Quality of English LanguageModerate editing of English language required